# ActAlign: Zero-Shot Fine-Grained Video Classification via Language-Guided Sequence Alignment

**Amir Aghdam**                                                    *amir.aghdam@temple.edu*
*Department of Computer Science, Temple University*
*Philadelphia, PA, USA*

**Vincent Tao Hu**                                                 *taohu620@gmail.com*
*CompVis @ LMU Munich, Munich Center for Machine Learning*
*Munich, Germany*

**Bjorn Ommer**                                                    *bommer@lmu.de*
*CompVis @ LMU Munich, Munich Center for Machine Learning*
*Munich, Germany*

**Reviewed on OpenReview:** *https://openreview.net/forum?id=Nwzn4qMTGb*

## Abstract

We address the task of zero-shot video classification for extremely fine-grained actions (e.g., Windmill Dunk in basketball), where no video examples or temporal annotations are available for unseen classes. While image–language models (e.g., CLIP, SigLIP) show strong open-set recognition, they lack temporal modeling needed for video understanding. We propose ActAlign, a *truly zero-shot*, training-free method that formulates video classification as a sequence alignment problem, preserving the generalization strength of pretrained image–language models. For each class, a large language model (LLM) generates an ordered sequence of sub-actions, which we align with video frames using Dynamic Time Warping (DTW) in a shared embedding space. Without any video–text supervision or fine-tuning, ActAlign achieves 30.4% accuracy on ActionAtlas—the most diverse benchmark of fine-grained actions across multiple sports—where human performance is only 61.6%. ActAlign outperforms billion-parameter video–language models while using ∼**8× fewer parameters**. Our approach is *model-agnostic and domain-general*, demonstrating that structured language priors combined with classical alignment methods can unlock the open-set recognition potential of image–language models for fine-grained video understanding.
Code Link: https://amir-aghdam.github.io/act-align/

## 1 Introduction

Understanding fine-grained human activities in video—such as distinguishing a *hook shot* from a *layup* in basketball, or recognizing tactical formations in football—requires parsing subtle, temporally ordered visual cues across frames. These fine-grained actions unfold in structured sequences of sub-actions and are often nearly indistinguishable from one another in appearance. In contrast to general activities like swimming, which can often be inferred from a single frame showing a person in water, fine-grained recognition demands attention to temporally extended object interactions, spatial relations, and high-level intent. As such, models must not only understand what is present in a video but also *when* and *how* key sub-actions occur. This requires accurately aligning the temporal progression of sub-actions with each fine-grained action to ensure a correct prediction, as shown in Figure 1. While their high-level label is barely recognized by action recognition models (e.g. such as *Roundhouse kick in Kickboxing*).

At the same time, contrastive vision–language models such as CLIP Radford et al. (2021) and SigLIP Zhai et al. (2023) have demonstrated impressive open-set fine-grained recognition in image domains by training

**Image-Text Embedding Space**

Figure 1: ActAlign improves zero-shot fine-grained action recognition by modeling them as structured language sequences. By aligning sub-action descriptions with video frames (green vs. red paths), we achieve more accurate predictions **without requiring any video-text training data.**

on massive image–text pairs. These models learn a shared image–text latent space, which allows both visual inputs and natural language descriptions to be embedded for direct comparison without task-specific supervision. This enables zero-shot classification using natural language prompts and has been widely adopted for downstream recognition tasks. However, extending these capabilities to video understanding introduces new challenges and requires temporal modeling. Existing methods that adapt CLIP-style models to video recognition either average frame-level features Rasheed et al. (2023); Zohra et al. (2025)—ignoring temporal structure—or fine-tune on target datasets Wang et al. (2022); Ni et al. (2022); Kim et al. (2024); Wang et al. (2024a), sacrificing generalization and open-set recognition. In both cases, the fine-grained temporal semantics of actions are lost or diluted.

Recent video–language architectures and instruction-tuned LLM-based systems such as Video-LLaMA Zhang et al. (2023), VideoChat Li et al. (2023) mPLUG-Owl Ye et al. (2023), Qwen2-VL Wang et al. (2024b), and DeepSeek-JanusPro Chen et al. (2025) enable open-ended, dialog-style video understanding through heavy instruction tuning, but they are not tailored for fine-grained video recognition.

Meanwhile, textual grounding (image–text alignment) remains a central challenge in interactive video-language models, especially for open-set and fine-grained video recognition. Dynamic Time Warping (DTW) Vintsyuk (1968), a classical algorithm for aligning temporally mismatched sequences, has seen renewed interest through differentiable variants Dogan et al. (2018); Chang et al. (2019) designed for supervised image–text temporal alignment. Yet, these methods rely either on annotated transcripts or example support videos or originally proposed for supervised training, making them impractical for zero-shot recognition. Likewise, approaches using part-level or attribute-level supervision Wu et al. (2023); Zhu et al. (2024) offer fine-grained cues but lack the ability to model the temporal structure between language-defined actions and visual content.

In this work, we introduce ActAlign, a novel framework that brings the open-set generalization power of image–text models to video classification of *extremely fine-grained actions* through language-guided sub-action alignment in a truly *zero-shot setting*. Rather than tuning a model for a specific domain or collapsing the video into a static representation, ActAlign operates in a training-free setting: for each unseen action class, we use a large language model (LLM) to generate a structured sequence of temporal sub-actions that semantically define the class. Then, using the pretrained SigLIP model Zhai et al. (2023) to extract frame-wise visual and sub-action features, we align the frame sequence with the LLM-generated sub-action script via Dynamic Time Warping (DTW) (see Figure 1). This allows us to compute a soft alignment score between different action classes that respects both content and temporal ordering, enabling fine-grained classification in a truly zero-shot manner.

**Our contributions are as follows:**

- We introduce a novel approach for zero-shot fine-grained video recognition that models each action as a *general, structured temporal sequence of sub-actions* derived solely from action names—**without access to videos or transcripts**.

- We propose ActAlign, a *domain-general, model-agnostic* and zero-shot framework that applies the open-set generalization strength of image–text models to the challenging task of fine-grained and zero-shot video classification by **reformulating the task to sequence matching** without requiring any video–text supervision.

- We demonstrate that ActAlign surpasses zero-shot and CLIP-based model baselines, outperforming even billion-parameter video–language models on the most challenging and diverse benchmark to our knowledge, ActionAtlas—where human-level accuracy caps at 61.64%.

## 2 Related Work

### 2.1 Action Recognition

Action recognition has been extensively studied, with early work leveraging hand-crafted features and classic classifiers Dalal & Triggs (2005); Jain et al. (2013); Wang & Schmid (2013). The field has since evolved to deep learning approaches Carreira & Zisserman (2017); Simonyan & Zisserman (2014); Tran et al. (2018); Wang et al. (2016), including CNNs, RNNs, and more recently, Transformer-based models Arnab et al. (2021); Bertasius et al. (2021); Kim et al. (2025); Yoshida et al. (2025), which model spatio-temporal dynamics more effectively. Fine-grained action recognition remains challenging due to subtle inter-class variations over time. Recent methods tackle this under supervised Leong et al. (2022); Sun et al. (2022), semi-supervised Li et al. (2022c); Dave et al. (2025), and few-shot Tang et al. (2023); Hong et al. (2021); Wang et al. (2021a) settings to mitigate annotation costs. Yet, they remain reliant on some level of fine-grained labeling and are constrained to specific domains.

### 2.2 Vision-Language Models

#### 2.2.1 Image–Language Models

Foundational image–language models such as CLIP Radford et al. (2021), SigLIP Zhai et al. (2023), and ALIGN Jia et al. (2021) learn joint image–text embeddings from large-scale image–caption pairs. Such modeling enables strong open-set recognition without task-specific supervision. These models are widely adopted as pretrained backbones for downstream tasks, including visual question answering Li et al. (2021); Tsimpoukelli et al. (2021); Li et al. (2022b); Steiner et al. (2024), image captioning and generation Mokady et al. (2021); Wang et al. (2021b), and few-/zero-shot classification Zhou et al. (2021); Khattak et al. (2025). However, they lack temporal modeling, which limits their open-set recognition capability to video inputs.

#### 2.2.2 Video–Language Models

Early efforts in video modeling focused on self-supervised pretraining. Models like ActBERT Zhu & Yang (2020) and VideoBERT Sun et al. (2019) applied masked language modeling to video frames. This enabled better transfer to video–language tasks through finetuning on paired video–text data Miech et al. (2020); Lei et al. (2021); Xu et al. (2021); Feichtenhofer et al. (2022); Li et al. (2022a). Recent progress in LLM reasoning has driven their integration with visual encoders, enabling open-ended video understanding via user prompts. These models Ye et al. (2023); Li et al. (2023); Zhang et al. (2024); Wang et al. (2024b), such as *Video-LLaMA* Zhang et al. (2023), combine pretrained visual backbones with chat-centric LLMs to generate spatio-temporal reasoning in conversational settings. While these models excel at open-ended question answering, they require extensive instruction tuning and are not optimized for fine-grained video recognition.

### 2.3 Sequence Alignment

Sequence alignment is a long-standing problem in fields such as biology and signal processing. A widely used technique for aligning temporally misaligned signals is Dynamic Time Warping (DTW)Berndt & Clifford (1994); Keogh & Pazzani (2000); Rath & Manmatha (2003), which computes the optimal warping path between two ordered sequences. Early work extended DTW to video tasksHuang et al. (2016); Richard et al. (2018). For instance, a seminal approach Bojanowski et al. (2015) proposed aligning videos to ordered scripts by enforcing the temporal order of events. With the rise of deep learning, DTW has been adapted into differentiable forms Dogan et al. (2018); Chang et al. (2019), enabling greater flexibility and improved learning capacity. A notable example is $OTAM$ Cao et al. (2020), which leverages DTW for representation learning in few-shot video classification by aligning each video to its support set. However, these methods often assume access to ground-truth transcripts for each video or rely on task-specific supervision, such as paired video–text exemplars.

**Our Work**  In contrast to prior methods that discard temporal structure, rely on task-specific supervision, or require support data for alignment, ActAlign introduces a language-guided sequence alignment framework for zero-shot fine-grained video classification.

## 3 Method

### 3.1 Problem Definition

Let $\mathcal{D} = (V_i, y_i)_{i=1}^N$ denote a dataset of $N$ videos, where each video $V_i$ is associated with a ground-truth fine-grained class label $y_i$ drawn from a set of $M$ candidate classes $\mathcal{Y} = c_1, \ldots, c_M$. Each video $V_i$ consists of a sequence of $T_i$ frames as defined in Eq. 1:

$$V_i = \{\mathbf{v}_i^t\}_{t=1}^{T_i}, \quad \mathbf{v}_i^t \in \mathbb{R}^{H \times W \times 3}, \tag{1}$$

where $H$ and $W$ denote the frame height and width. In our zero-shot setting, no video examples of the target classes $\mathcal{Y}$ are used for training or tuning; only high-level action names $c_j$ are provided. The goal is to construct a function $f : \mathcal{V} \times \mathcal{Y} \to \mathbb{R}$ that effectively maps the sequence of video frames into their correct action class $y$. The predicted class label $\hat{y}_i$ for a video $V_i$ is given by Eq. 2:

$$\hat{y}_i = \arg \max_{c_m \in \mathcal{Y}} f(V_i, c_m). \tag{2}$$

To leverage semantic priors from LLMs, we automatically decompose each class label $c_m$ into an ordered, variable-length sequence of $K_m$ textual sub-actions, as defined in Eq. 3:

$$S_m = [s_{m,1}, s_{m,2}, \ldots, s_{m,k_m}], \tag{3}$$

where $s_{m,k}$ is a concise natural-language description of the $k$-th step in executing action $m$.

### 3.2 Our Method

We define $f(V_i, c_m)$ as the alignment score between the visual frame embeddings $\{\mathbf{v}_i^t\}$ and the sub-action sequence $S_m$, computed via Dynamic Time Warping (DTW). This alignment is performed in the image–text embedding space, without requiring any fine-tuning or video examples from the target label set. Figure 2 illustrates the pipeline of our proposed approach.

### 3.3 Preliminary Sub-action Generation by LLM

In domains requiring fine-grained distinctions—such as differentiating tactical plays in sports—the high-level action class name $c_m$ often lacks sufficient discriminatory power. To address this, we define a mapping

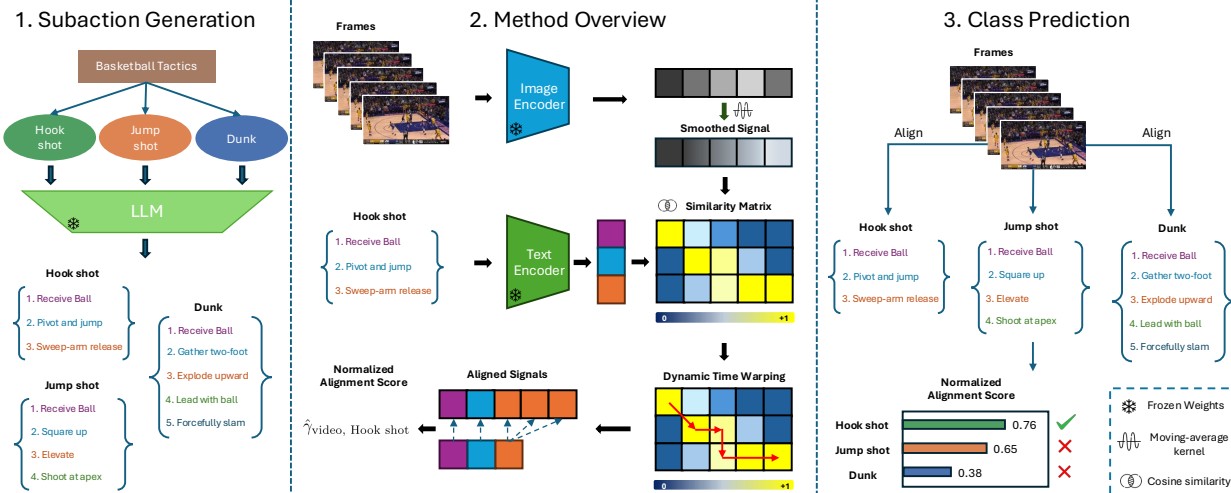

Figure 2: **Our ActAlign Method Overview.** (**1**) *Sub-action Generation:* Given fine-grained actions (e.g. Basketball Tactics), we prompt an LLM to decompose each action (e.g. Hookshot, JumpShot, Dunk) into a temporal sequence of sub-actions. (**2**) *Temporal Alignment:* Video frames are encoded by a frozen pretrained vision encoder and smoothed via a moving-average filter. Simultaneously, each sub-action is encoded by the text encoder. We compute a cosine-similarity matrix between frame and sub-action embeddings, then apply Dynamic Time Warping (DTW) to find the optimal alignment path and normalized alignment score. (**3**) *Class Prediction:* We repeat this process for each candidate action $m$, compare normalized alignment scores $\hat{\gamma}_{\text{video},m}$, and select the action sequence with the highest score as the final prediction.

$\mathcal{P} : \mathcal{Y} \to \mathcal{S}$, where $\mathcal{S}$ denotes the space of ordered sub-action scripts. For each class $c_m \in \mathcal{Y}$, we generate a sequence $S_m = [s_{m,1}, \ldots, s_{m,K_m}] = \mathcal{P}(c_m)$, where each $s_{m,k}$ describes a trackable and temporally ordered sub-action.

We instantiate $\mathcal{P}$ using GPT-4o via carefully constructed natural language prompts. Given a set of candidate classes $\{c_1, \ldots, c_M\}$, the LLM is instructed to:

1. Decompose each $c_m$ into a variable- or fixed-length sequence of semantically coherent sub-actions.

2. Return the list $[s_{m,1}, \ldots, s_{m,K_m}]$ in a consistent, structured format.

By leveraging the LLM's extensive prior knowledge, we obtain sub-action sequences without any manual labels or video supervision. These sequences serve as semantic reference signals for temporal alignment.

**Importance of Context in Sub-Actions.** Terse descriptions (e.g., "drive forward") are often ambiguous and may refer to multiple sports, which hampers DTW alignment due to poor textual grounding. In contrast, context-rich prompts (e.g., "player drives forward to the rim") produce more discriminative sub-actions that are reliably grounded with visual evidence.

**Context Augmentation** We enrich each sub-action by embedding it into a contextualized template: `This is a video of doing <action name> in <sport domain> with <sub-action>`. This substantially improves semantic clarity and further disambiguates sub-actions. The sport domain is automatically provided in the ActionAtlas metadata (and could also be inferred from the action name itself).

## 3.4 Visual and Semantic Feature Encoding

Once each class label $c_m$ is decomposed into its sub-action sequence $S_m$, we project both video frames and sub-actions into a shared $d$-dimensional embedding space using the pre-trained SigLIP image–text model, which is recognized for its strong zero-shot recognition performance.

**Visual Embeddings**   Let $\phi_v : \mathbb{R}^{H \times W \times 3} \to \mathbb{R}^d$ denote the vision encoder. For video $V_i = \{\mathbf{v}_i^t\}_{t=1}^{T_i}$, we compute frame-level embeddings:

$$\mathbf{z}_i^t = \phi_v(\mathbf{v}_i^t) \in \mathbb{R}^d, \quad t = 1, \dots, T_i. \tag{4}$$

Stacking yields $Z_i = [\mathbf{z}_i^1, \dots, \mathbf{z}_i^{T_i}] \in \mathbb{R}^{d \times T_i}$.

**Semantic Embeddings**   Let $\phi_t : \mathcal{T} \to \mathbb{R}^d$ be the text encoder. For each class $c_m$ and its sub-action sequence $S_m$, we embed each step:

$$\mathbf{u}_{m,k} = \phi_t(s_{m,k}) \in \mathbb{R}^d, \quad k = 1, \dots, K_m, \tag{5}$$

stitching into $U_m = [\mathbf{u}_{m,1}, \dots, \mathbf{u}_{m,K_m}] \in \mathbb{R}^{d \times K_m}$.

**Shared Latent Space**   We normalize all embeddings so that the similarity

$$\text{sim}(\mathbf{z}_i^t, \mathbf{u}_{m,k}) = \frac{\mathbf{z}_i^{t\top} \mathbf{u}_{m,k}}{|\mathbf{z}_i^t| \cdot |\mathbf{u}_{m,k}|} \tag{6}$$

is a cosine similarity measure between frame $\mathbf{v}_i^t$ and sub-action $s_{m,k}$ of class $c_m$ (see Eq. 6). This cross-modality similarity forms the basis for alignment in the next step.

## 3.5   Dynamic Time Warping

After feature encoding, each video yields a visual embedding sequence $Z_i$ and ordered sequences of sub-action embeddings $U_m$ for each class $c_m$. We treat $U_m$ as the reference semantic signal and $\widetilde{Z}_i$ as the query visual signal. The $U_m$ could also be viewed as a prototype sequence for class $c_m$.

**Signal Smoothing**   Real-world footage often contains abrupt scene changes or irrelevant frames (e.g., replays, advertisements) that introduce noise into $Z_i$. To mitigate this, we apply a simple, parameter-free 1D moving-average filter of width $w$ across the temporal dimension:

$$\widetilde{\mathbf{z}}_i^t = \frac{1}{w} \sum_{\tau = t - \lfloor w/2 \rfloor}^{t + \lfloor w/2 \rfloor} \mathbf{z}_i^\tau, \tag{7}$$

with boundary conditions handled via zero padding. The kernel width $w$ controls the trade-off between noise reduction and temporal resolution. The smoothing effectively aggregates short-term temporal context, dampening isolated spikes while preserving the overall action dynamics. As it introduces no learnable parameters or additional training, it remains lightweight and fully compatible with the training-free setting.

**Affinity Matrix Construction**   Let the smoothed visual embeddings for video $V_i$ be $\widetilde{Z}_i = [\widetilde{\mathbf{z}}_i^1, \dots, \widetilde{\mathbf{z}}_i^{T_i}]$ and the sub-action sequence embeddings for class $c_m$ be $U_m = [\mathbf{u}_{m,1}, \dots, \mathbf{u}_{m,K_m}]$. We first compute the raw cosine similarity matrix:

$$A_{k,t}^{(m,i)} = \left\langle \mathbf{u}_{m,k}, \widetilde{\mathbf{z}}_i^t \right\rangle, \quad A^{(m,i)} \in \mathbb{R}^{K_m \times T_i} \tag{8}$$

where each $\langle \cdot, \cdot \rangle$ is the inner product of $\mathcal{L}2$-normalized vectors, yielding values in $[-1, 1]$. Following the SigLIP prediction approach, we then apply a `sigmoid` function $\sigma(\cdot)$ to transform these values into affinity scores in $[0, 1]$:

$$\hat{A}_{k,t}^{(m,i)} = \sigma\big(\alpha\, A_{k,t}^{(m,i)} + \beta\big), \tag{9}$$

where $\alpha, \beta$ are pre-learned scaling parameters as part of SigLIP's pretraining recipe. The resulting $\hat{A}^{(i,m)}$ is used as the input affinity matrix for DTW alignment.

**DTW Alignment and Scoring**  Given the affinity matrix $\hat{A}^{(m,i)}$ for class $c_m$ and video $V_i$ (defined in Eq. 9), we seek a warping path $W^{(m,i)} = \{(k_1, t_1), \ldots, (k_L, t_L)\}$ that maximizes cumulative similarity under monotonicity and continuity constraints:

$$W^{(m,i)} = \arg\max_{W} \sum_{(k,t)\in W} \hat{A}^{(m,i)}_{k,t},$$

$$\text{s.t.} \quad W \text{ is a valid warping path between } [1, K_m] \text{ and } [1, T_i]. \tag{10}$$

We solve this using dynamic programming by selecting the path with highest alignment (i.e. similarity value) at each step:

$$D_{k,t} = \hat{A}^{(m,i)}_{k,t} + \max\{D_{k,t-1}, D_{k-1,t}, D_{k-1,t-1}\}, \tag{11}$$

with the base case $D_{0,*} = D_{*,0} = -\infty$. The final alignment score is $\max_{k,t} D_{k,t}$, and backtracking recovers the optimal warping path $W^{(m,i)}$ for sub-action sequence of class $c_m$ and video $V_i$.

**Prediction**  We compute the raw alignment score, as defined in Eq. 12, as the sum of the similarity values of the optimal warping path $W^{(m,i)}$ obtained in Eq. 10.

$$\gamma_{i,m} = \sum_{(k,t)\in W^{(m,i)}} \hat{A}^{(m,i)}_{k,t}, \tag{12}$$

$\hat{A}^{(m,i)}$ is the affinity matrix introduced in Eq. 9. To mitigate the bias toward longer warping paths (which can accumulate higher raw scores), we normalize $\gamma_{i,m}$ by the path length, resulting in the average alignment score:

$$\hat{\gamma}_{i,m} = \frac{1}{|W^{(m,i)}|}\gamma_{i,m}, \tag{13}$$

where $|W^{(m,i)}|$ is the number of matched frame–sub-action pairs. Since the similarity values in $\hat{A}^{(i,m)}$ lie in $[0, 1]$ (due to the sigmoid in Eq. 9), the normalized alignment score $\hat{\gamma}_{i,m}$ also lies in the range $[0, 1]$.

Finally, we predict the class whose sub-action sequence best aligns—on average—with the observed video frames. This is done by selecting the class with the highest normalized alignment score:

$$\hat{y}_i = \arg\max_{c_m \in \mathcal{Y}} \hat{\gamma}_{i,m}. \tag{14}$$

## 4   Experiment

### 4.1   Experimental Setup

**Dataset**  Our approach is dataset- and domain-agnostic, as it relies solely on language-driven sub-action generation from LLMs and cross-modal alignment without task-specific supervision. To rigorously evaluate its generality, we adopt ActionAtlas Salehi et al. (2024)—to our knowledge, the most diverse and challenging benchmark for fine-grained action recognition across various domains. For each video $V_i$, we retain its multiple-choice candidate set $\{c_{i,1}, \ldots, c_{i,M_i}\}$, and replace each class label with an LLM-generated sub-action sequence. (Dataset statistics is provided in the Appendix.)

| Method | #Param | Top-1 (%) ↑ | Top-2 (%) ↑ | Top-3 (%) ↑ |
|---|---|---|---|---|
| Random (10 Trials) | - | 20.81 | 42.04 | 62.50 |
| Human Evaluation (Oracle) | - | 61.64 | - | - |
| mPLUG-Owl-Video Ye et al. (2023) | 7B | 19.49 | - | - |
| VideoChat2 Li et al. (2023) | 7B | 21.27 | - | - |
| VideoLLaMA Zhang et al. (2023) | 8B | 22.71 | - | - |
| LLaVA-Next-Video Zhang et al. (2024) | 7B | 22.90 | - | - |
| Qwen2–VL Wang et al. (2024b) | 7B | 30.24 | - | - |
| X-CLIP-L/14-16F Ni et al. (2022) | 0.6B | 16.26 | 33.74 | 49.89 |
| SigLIP–so400m Zhai et al. (2023) (mean-pool) | 0.9B | 22.94 | 42.20 | 63.70 |
| + DTW Alignment (*Ours*) | 0.9B | 25.72 | 45.99 | 66.26 |
| **ActAlign (*Ours*)** | 0.9B | **30.40 ± 0.11** | **53.01** | **70.27** |

Table 1: **Zero-shot classification results on ActionAtlas.** Our method achieves state-of-the-art Top-1, Top-2, and Top-3 accuracy, outperforming all baselines and billion-parameter video–language models without any video–text supervision. These results highlight the open-set recognition capability of image-text models and the effectiveness of structured sub-action alignment over flat representations such as mean-pooling. We use context-rich prompting strategy.

**Evaluation Metrics**   Following Chen & Huang (2021); Bosetti et al. (2025), we report Top-$k$ accuracy for $k \in \{1, 2, 3\}$:

$$\text{Top-}k = \frac{1}{N} \sum_{i=1}^{N} \mathbb{I}\Big(\text{rank}_i(\hat{y}_i) \leq k\Big),$$

where $\mathbb{I}$ is the indicator function and $\text{rank}_i(\hat{y}_i)$ is the position of the ground-truth label in the descending list of scores $\{\hat{\gamma}_{i,1}, \ldots, \hat{\gamma}_{i,M_i}\}$. This accounts for typical cases where fine-grained actions are semantically similar and alignment scores are closely clustered, allowing improvement to be captured within a narrowed candidate set.

**Experimental Detail**   We use SigLIP–so400m (patch size 14, $d = 384$, 878M parameters). We apply a moving-average smoothing with a fixed window size of $w = 30$ frames (1s @30 fps) to reduce transient noise and emphasize consistent motion patterns. All experiments run on a single NVIDIA RTX A5000 GPU (25 GB). Inference consists of embedding extraction followed by DTW alignment over $N$ class candidates, each with $M$ sub-actions, and $T$ video frames—yielding $\mathcal{O}(NMT)$ time. Following the zero-shot protocol, no example videos or sub-action sequence from these classes are used for any training or tuning. We only use candidate action class names $c_m$ to prompt for sub-action generation.

## 4.2   Experimental Result

**Zero-Shot Comparisons**   We begin with a random choice baseline and a SigLIP zero-shot baseline using mean-pooled frame embeddings. Following ViFi-CLIP Rasheed et al. (2023), each video $V_i$ is encoded as $\bar{\mathbf{z}}_i = \frac{1}{T_i} \sum_{t=1}^{T_i} \mathbf{z}_i^t$, where mean-pooled embeddings $\bar{\mathbf{z}}$ is compared via cosine similarity to each name embedding $\phi_t(c_j)$.

We further compare against open-source video–language models, including entries from the ActionAtlas leaderboard and fine-tuned CLIP variants. Despite using no video–text supervision, our method outperforms the SigLIP baseline by +7% Top-1 and +11% Top-2 accuracy, surpassing all baselines and billion-parameter models with ∼8× fewer parameters. These larger models are often optimized for interactive tasks through extensive instruction tuning, rather than for classifying fine-grained actions. In contrast, image–language models are trained on large-scale image–text pairs, making them particularly well-suited for open-set recognition when paired with temporal alignment mechanisms.

As shown in Table 1, these gains stem from our subaction-level alignment, which enables more discriminative and interpretable classification than global frame pooling method.

| Configuration | Top-1 (%) ↑ | Top-2 (%) ↑ | Top-3 (%) ↑ |
|---|---|---|---|
| SigLIP Zhai et al. (2023) (mean-pool) | 22.94 | 42.20 | 63.70 |
| + DTW Alignment | 25.72 | 45.99 | 66.26 |
| + Context Augmentation | 30.07 | 52.67 | 70.49 |
| + Signal Smoothing | 30.29 | 53.01 | 70.27 |

Table 2: **Ablation results of various design** under context-rich prompting. DTW alignment introduces temporal matching, context augmentation reduces sub-action ambiguity, and signal smoothing mitigates frame-level noise.

| Configuration | Top-1 (%) ↑ | Top-2 (%) ↑ | Top-3 (%) ↑ |
|---|---|---|---|
| SigLIP (mean-pool) | 22.94 | 42.20 | 63.70 |
| SigLIP (mean-pool) w/ Context Augmentation | 28.51 | 50.89 | 70.38 |
| SigLIP (mean-pool) w/ Bag-of-Words | 29.06 | 51.34 | 69.49 |
| *ActAlign (ours)* | 30.29 | 53.01 | 70.27 |

Table 3: **Comparison with baselines under context-rich prompting.** Context augmentation enriches *action names* (as opposed to sub-actions in Table 2) with domain names, while bag-of-words mean-pools sub-action descriptions into a unified representation.

**Ablation Studies**  We ablate each component of ActAlign on ActionAtlas (Table 2), starting from a mean-pooled SigLIP baseline. Adding DTW alignment introduces temporal structure and yields consistent gains. Context augmentation—injecting domain context (e.g., "Sprint start" → "Sprint start in basketball")—produces the largest boost by resolving semantic ambiguity. Signal smoothing offers a modest but complementary improvement by reducing frame-level noise and clarifying action boundaries.

**Baseline Comparisons**  To further evaluate the generalization power of SigLIP without any temporal information, we evaluate SigLIP performance by the similar context augmentation technique we used in our framework and with mean-pooling the bag of sub-actions and comparing the mean-pooled representations with video representation (See Table 3). In all of these baselines, video frame representations are mean-pooled. We further investigate classification performance under randomized and reversed sub-action orders at Table 4

**Sub-action Baselines**  To evaluate the impact of sub-actions on alignment, we test DTW alignment performance under two perturbations: randomizing sub-actions and reversing their order on short-fixed sub-actions. Results are shown in Table 4.

We observe that the upper bound on performance is closely tied to the specificity and coherence of LLM-generated sub-action sequences.

**Prompt Variations**  We evaluate two prompt strategies for generating sub-action scripts using GPT-4o, keeping all other components fixed:

| Configuration | Top-1 (%) ↑ | Top-2 (%) ↑ | Top-3 (%) ↑ |
|---|---|---|---|
| Reversed Order | 26.39 | 47.66 | 67.04 |
| Randomized Order (5 Trials) | $26.63 \pm 0.14$ | $47.05 \pm 0.29$ | $67.00 \pm 0.25$ |
| Normal | 27.06 | 46.88 | 66.82 |

Table 4: **Comparison with sub-action baselines.** We evaluate classification performance by randomizing and reversing the order of *short-fixed* sub-actions due to their larger sequence size.

| Prompt | Description | Top-1 (%) ↑ |
|--------|-------------|-------------|
| Short-fixed | 2-word, fixed 10 sub-actions | 27.06 |
| Context-rich | context-rich, variable-length | **30.29** |

Table 5: **Impact of prompt strategy.** Context-rich prompting improves zero-shot classification performance by producing more specific and informative sub-actions.

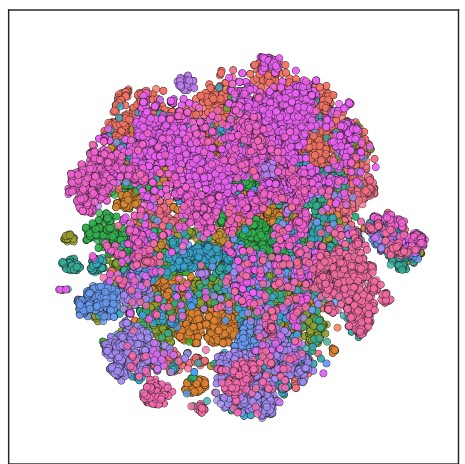 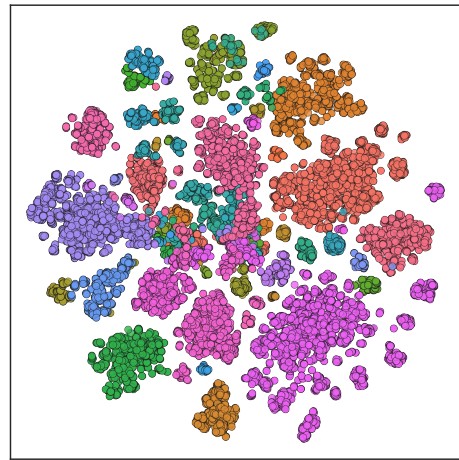


w/o Context Augmentation     w/ Context Augmentation


Figure 3: **t-SNE visualization of sub-action embeddings.** Each color corresponds to a sport domain. Augmenting sub-actions with context yields more discriminative clusters and improves textual grounding.

- **Short-Fixed:** Prompts GPT-4o to generate exactly 10 terse (2–3 word) sub-actions per class using a fixed structure.

- **Context-Rich:** Produces variable-length, context-rich sub-action scripts incorporating domain-specific cues (e.g., "Wrestler", "Rider").

Table 5 shows the performance of each strategy. The context-rich prompt with domain-specific context achieve the highest accuracy. In contrast, short-fixed prompts—lacking sufficient semantic specificity—perform worst. These results highlight that reducing ambiguity in sub-action descriptions directly improves alignment quality and classification performance.

**Sub-action Embedding Structure** Figure 3 shows 2D t-SNE projections of sub-action embeddings with and without context augmentation under context-rich prompts. Adding domain-specific cues (`<sub-action>` `in <sport name>`) results in tighter, more coherent clusters—indicating better semantic structure and separation.

**Sub-action Sequence Examples** Table 6 shows LLM-generated sub-action sequences for two *Figure skating* tactics under our context-rich prompting. The scripts highlight ordered, salient steps enabling precise temporal matching.

**Alignment Heatmaps and Paths** Figure 4 visualizes the cosine similarity matrix and DTW path for a correctly classified action (right) and an incorrect candidate (left). The correct sequence yields high-similarity regions with a monotonic path. In contrast, the incorrect script shows sparse similarity regions, with the DTW path forced to follow the single most similar alignment trace.

**Signal Smoothing.** Figure 5 compares similarity matrices before and after applying a moving-average filter. Without smoothing, rapid scene changes introduce noise and scattered peaks. Smoothing yields cleaner similarity surfaces with clearer action boundaries.

---

**Sub-action Script**

---

*Biellmann Spin*

1. Begins upright spin on one foot with arms extended and free leg behind, 2. Gradually pulls free leg upward behind the back using both hands, 3. Raises the free leg above head level while arching the back dramatically, 4. Grasps the blade of the free skate with both hands overhead, 5. Extends spinning leg vertically while maintaining centered spin on skating foot, 6. Maintains high-speed rotation with body in extreme vertical split position

---

*Flying Camel Spin*

1. Skater glides forward with arms extended and knees bent in preparation, 2. Performs a powerful jump off the toe pick while swinging free leg upward, 3. Rotates mid-air with body extended horizontally like a 'T' shape, 4. Lands on one foot directly into a camel spin position with torso parallel to ice, 5. Extends free leg backward and arms outward while spinning on the skating leg, 6. Maintains fast, centered rotation in the horizontal camel position

---

Table 6: **LLM-generated sub-action scripts for figure skating tactics.** Shown for the *Biellmann Spin* and *Flying Camel Spin* examples in Figure 4, these sequences are generated using context-rich prompting and provide semantically detailed, temporally ordered steps for alignment in our zero-shot framework.

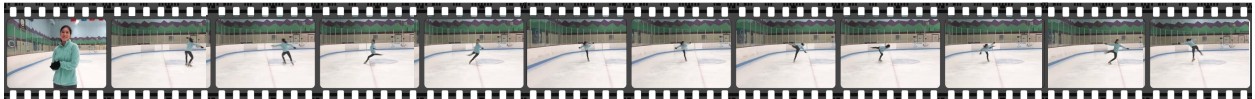

(a) **Video:** Performing *flying camel spin* in figure skating.

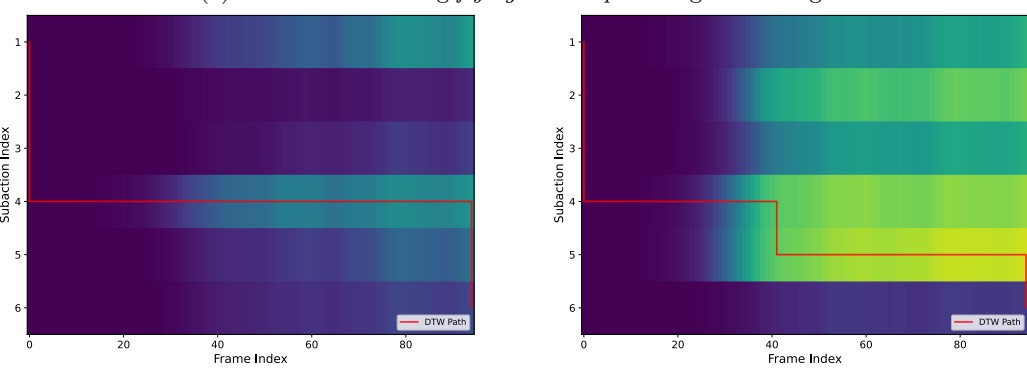

(b) Biellmann spin ✗           (c) Flying camel spin ✓

Figure 4: **DTW alignment paths** for an incorrect prediction (left) versus a correct classification (right). The correct class exhibits clearer segmentation and higher alignment quality. The sub-action scripts are provided in Table 6.

**Failure Case Analysis** We observe two primary failure modes in our framework:

- **Ambiguous sub-actions:** Vague LLM-generated steps (e.g., "move to position") result in sparse similarity matrices, weakening DTW's ability to discriminate between candidates. As shown in Figure 6, context-rich prompts produce clearer alignment regions than short-fixed ones.

- **Global alignment bias:** Vanilla DTW enforces full-sequence alignment, which is suboptimal when actions start mid-clip or exhibit temporal shifts. Without a local alignment mechanism, early or trailing sub-actions can introduce noise. Figure 4 shows a temporal shift case that does not affect prediction here but may cause errors in more competitive settings.

These limitations underscore the importance of precise sub-action design and motivate future improvements in alignment robustness and prototype refinement.

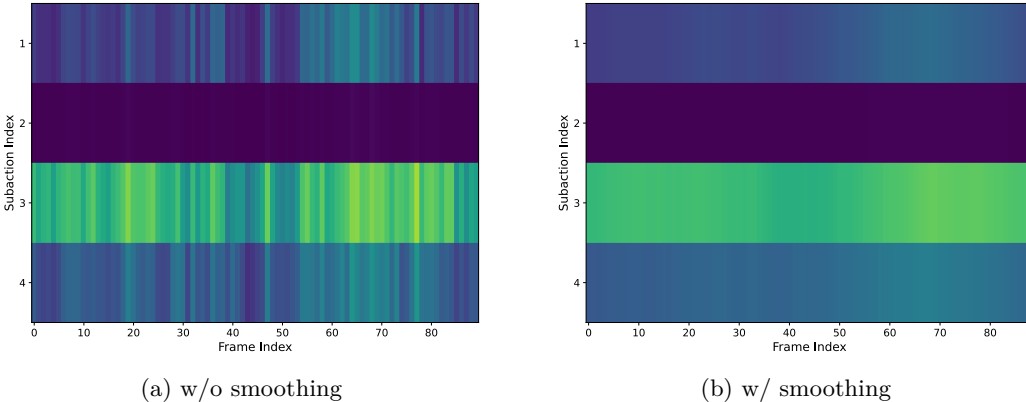

(a) w/o smoothing          (b) w/ smoothing

Figure 5: **Signal smoothing** reduces high-frequency noise and enhances transition between sub-actions. Similarity matrices before and after applying a moving-average filter ($w = 30$).

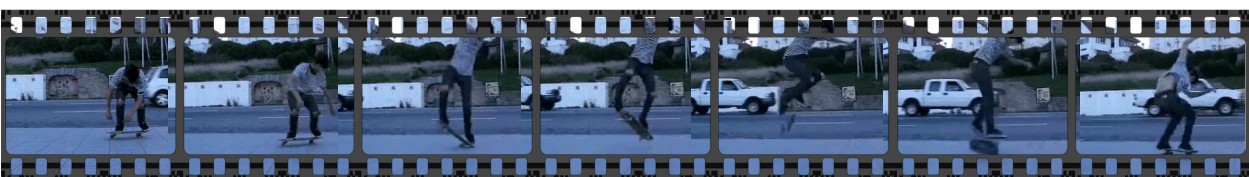

(a) **Video:** *Varial kickflip underflip body varial* in Skateboarding.

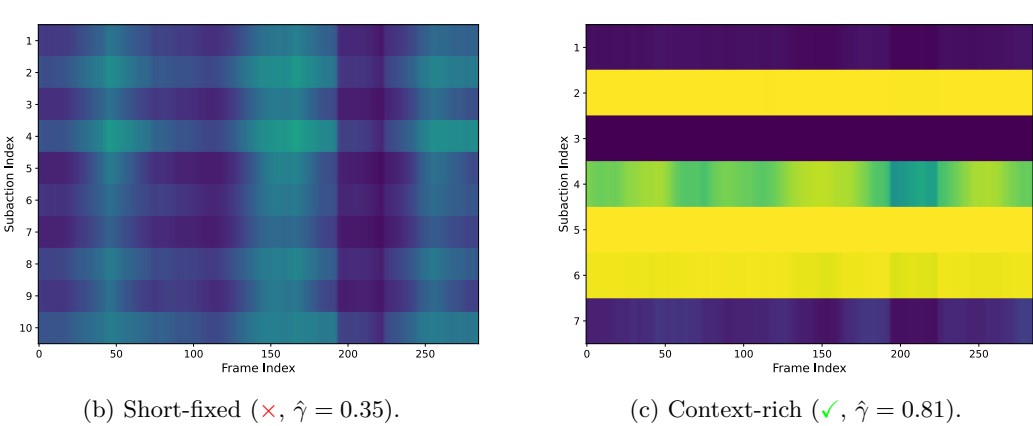

(b) Short-fixed ($\times$, $\hat{\gamma} = 0.35$).          (c) Context-rich ($\checkmark$, $\hat{\gamma} = 0.81$).

Figure 6: **Failure case** due to vague sub-actions for the action *Varial kickflip underflip body varial*. The normalized alignment score $\hat{\gamma}$ reflects overall sequence similarity. (b) The Short-fixed prompt yields a noisy similarity map and incorrect prediction. (c) The Context-rich prompt produces clearer alignment and correct classification. See Appendix for the detailed scripts.

# 5 Conclusion

We show that contrastive image–language models establish a surprisingly strong baseline for zero-shot fine-grained video classification, even when used with simple mean-pooling. To fully leverage this capability in extremely fine-grained settings, we propose **ActAlign**, a novel *zero-shot, domain-general, and model-agnostic* framework that revisits the classic Dynamic Time Warping (DTW) algorithm to cast video classification as a sequence-alignment problem. By aligning video frames with LLM-generated sub-action scripts, ActAlign introduces temporal structure into contrastive models without requiring any video–text training or fine-tuning. Evaluated on the highly challenging and diverse ActionAtlas Salehi et al. (2024) benchmark, our method achieves state-of-the-art performance, outperforming both CLIP-style baselines and billion-parameter video–language models.

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
