# OpenReview forum: "ActAlign: Zero-Shot Fine-Grained Video Classification via Language-Guided Sequence Alignment"
_TMLR — Accepted by TMLR_

### Review · Reviewer_pgw3 · 2025-08-28

**Summary Of Contributions:**

The paper proposes ActAlign, a training‑free, zero‑shot method for fine‑grained video classification that casts recognition as language‑guided sequence alignment. For each candidate class, a LLM decomposes the class name into an ordered list of sub‑actions; frames and sub‑actions are embedded with a frozen image–text model (SigLIP), and the method aligns the two sequences using Dynamic Time Warping (DTW). The final score is the mean similarity along the optimal warping path, and the predicted class is the one with the highest normalized alignment score. On ActionAtlas, ActAlign reports 30.29 Top‑1 vs 22.94 for SigLIP mean‑pooling and competitive/better results than several 7–8B video‑language models while using 0.9B image–text parameters. Figures on pp. 2 and 5 visualize the idea and pipeline; Tables 2–3 ablate DTW, context augmentation, and smoothing

Pro:
- Casting recognition as alignment in a pretrained image–text space is elegant and easy to reproduce. No video–text training or tuning is required
- The alignment path and sub‑action scripts make predictions inspectable in a way most VL baselines are not
- The qualitative heatmaps (Figures 5–6) help diagnose behavior

Con:
- The strongest competing VLM (Qwen2‑VL) is essentially tied (30.24 vs 30.29 Top‑1), and several VLM numbers are reported only for Top‑1 (Table 1). It’s not fully clear that prompting/decoding for these large models is optimized under the multiple‑choice setting used here
- The largest gain in Table 2 comes from context augmentation of the text (30.07 vs 25.72 after DTW). It is hard to isolate how much improvement is due to adding richer text vs DTW ordering
- Eq. (9) applies a sigmoid with “learned” $\alpha,\beta$, yet the method is otherwise training‑free. It is unclear whether these are fixed from SigLIP pretraining, hand‑set, or fitted on ActionAtlas.

**Audience:**

Yes

**Audience Explanation:**

This topic is clearly interesting to the community

**Claims And Evidence:**

Yes

**Claims Explanation:**

Table 1 (Section 4.2, p.7) shows ActAlign at 30.29 Top‑1 on ActionAtlas vs 22.94 for SigLIP mean‑pooling, with corresponding Top‑2/Top‑3 improvements. ActAlign (0.9B params) is on par with or slightly above Qwen2‑VL‑7B (30.24 Top‑1) and clearly above several other 7–8B models. Table 2 (p.8) starts from the mean‑pool baseline and adds components stepwise: +DTW (+2.8 Top‑1), +context augmentation (+4.35), +smoothing (+0.22).

**Requested Changes:**

1. Please spell out the exact prompts and decoding for each VLM, whether the models are constrained to choose among the multiple‑choice options, how many frames/tokens were fed, and whether audio was used.
2. The baseline Mean‑pooled SigLIP with the same context template
3. The baseline Bag‑of‑subactions that averages sub‑action text embeddings and compares to mean‑pooled frames.
4. Add a “randomly permuted sub‑action order” and a “reversed order” control
5. Report variance across multiple seeds (or LLM samples) for Top‑1; include one qualitative example where different phrasings lead to different alignments

---

> ### Author Response · Authors · 2025-09-29
> **Response to Reviewer pgw3**
>
> We sincerely thank the reviewer for their constructive and detailed feedback, and for recognizing the novelty and reproducibility of our alignment-based formulation. Below, we address each point in detail.
>
> **1. VLM Evaluation Settings and Reporting**
>
> We agree that evaluation details are essential for interpreting Table 1. Our reporting **strictly follows the official ActionAtlas[1] evaluation protocol and leaderboard** for open-source video–language models. The benchmark provides **multiple-choice question–answer pairs**, and each model is constrained to select from the given candidate actions.
>
> As part of the standardized preprocessing pipeline defined by ActionAtlas[1], **all audio signals, speech transcripts, and any textual cues visible in the video are discarded or blurred** before inference. This ensures that performance reflects only visual understanding rather than reliance on auxiliary textual or audio information.
>
> Furthermore, the official ActionAtlas paper provides a comprehensive analysis of how performance varies with different frame rates. The results we report here follow their leaderboard and correspond to the ++best-performing frame-rate setting for each model++ as reported in the original benchmark. The table below (reproduced from the ActionAtlas public leaderboard) summarizes the conditions:
>
> | Model | Frames | Tokens | Top-1 Acc. (%) |
> |-------|--------|--------|----------------|
> | Qwen2-VL-7B | 16 | 8 × 576 | 30.24 |
> | VideoLLaMA | 16 | 16 × 256 | 22.71 |
> | VideoChat2 | 64 | 64 × 196 | 21.27 |
> | mPLUG-Owl-Video | 16 | 16 × 256 | 19.49 |
> | LLaVA-Next-Video-7B | 64 | 64 × 144 | 22.90 |
>
>
> **2. Additional Baselines and Ablations**
>
> We have incorporated all requested additional baselines to isolate the contributions of each component:
>
> - **Mean-pooled SigLIP with context:** This baseline uses the context-augmented text without any additional alignment steps.
> - **Bag-of-subactions:** Here, sub-action embeddings are averaged and compared to mean-pooled frames.
> - **Randomized and Reversed sub-action order:** Here, we report classification performance under randomized and reversed sub-action orders.
>
> Results confirm that DTW and sequential ordering contribute additional gains beyond text enrichment; Bag-of-sub-actions control demonstrates that simple aggregation underperforms compared to our alignment-based formulation. While, ranomized or reversed-order sub-actions slightly under-performs the normal sequence.
>
> We acknowledge that the reviewer raised a very good question here as this experiment reveals subtle effects that warrant deeper investigation. We provide initial analysis and commit to conducting a more comprehensive study of ordering sensitivity in future work, which could be further improved by exploring more advanced prompt engineering beyond the basic prompting used in this version.
>
> **3. Sigmoid Scaling Parameters**
> We thank the reviewer for pointing out the ambiguity around the sigmoid scaling in Eq. (9). We clarify that these scaling coefficients are not learned or fine-tuned on ActionAtlas. Instead, they are part of SigLIP's pretraining recipe and derived from SigLIP’s pretrained model. No gradient updates are performed, and the method remains fully training-free across all domains. We included the description below Eq 9 to clarify that.
>
>
> **4. Variance, Prompt Robustness, and Qualitative Examples**
> We have extended our robustness analysis as requested:
> - We now report variance across three random seeds (or LLM samples), showing stable performance with only minor fluctuations in Top-1 accuracy. Overall, repeated LLM sampling does not lead to significant performance differences. However, we observe that the method could potentially be significantly improved by exploring more advanced prompting strategies.
> - We added a qualitative example illustrating how two different prompt phrasings generate distinct sub-action sequences and how these yield different alignment paths. In general, we note that DTW tends to find the few highly similar sub-actions within each sequence variant and explot them to maximize similarity.
>
>
> References:
> [1]. Mohammadreza Salehi, Jae Sung Park, Aditya Kusupati, Ranjay Krishna, Yejin Choi, Hannaneh Hajishirzi, and Ali Farhadi. ActionAtlas: A VideoQA Benchmark for Domain-specialized Action Recognition. Advances in Neural Information Processing Systems (NeurIPS) Datasets and Benchmarks Track, 2024. Available at: https://openreview.net/forum?id=6kc6Hdyknx

---

### Review · Reviewer_Nrv2 · 2025-09-04

**Summary Of Contributions:**

this paper proposes a new method ActAlign which is a training-free, zero-shot framework that re-purposes fine-grained video recognition as sequence alignment between an LLM-generated "expansion" or script of temporally ordered sub-actions (outputted only from the action name used in teh prompt) and framewise visual embeddings. It uses a pretrained image–text model (SigLIP) to embed frames and sub-actions and applies Dynamic Time Warping to score alignments—requiring no video–text supervision and remaining domain-general/model-agnostic. The empirical results are interesting and the ablation studies seem adequate.

**Audience:**

Yes

**Audience Explanation:**

The paper is about multimodal embeddings and zeroshot video/text alignment for automatic subaction annotation; beyond the main algorithm even the techniques are interesting and should interest the community.

**Claims And Evidence:**

Yes

**Claims Explanation:**

The empirical evidence is clear. The authors have agreed to release the code too.

**Requested Changes:**

This is not my field of research, and the experimental results are pleasantly suprising to me. My main concern (not really a concern, but more of a surprise) is how the embeddings generated through SigLIP (\phi_v, \phi_t) somehow embed the video and the text in the same domain so as that the simple dot product works. I understand the semantic expansion of a subclass into temporal subactions by LLMs intuitively, and I can see how a pre-trained LLM can handle this easily (thereby allowing the “zero-shot” no-explicit labelling effort). But is there a reason that the embeddings of vector and text outputted by SigLIP should be able to embed them in the “same” semantic space ? Is this something known from before ?

Did the authors think of any ablation experiments for different sequence of subactions generated by the same LLM through repeated or slightly different prompts? Would the (difference in) lengths of these subaction-sequences have an impact? I realize the authors have mentioned ablation studies and in light of expanded contextual information the impact of initial sequences that were generated may not be big.

The manuscript could benefit from additional details such as SigLIP and why it should embed video frames and text in the same frame? Similarly additional details on how signal smoothing was done, and how contextual expansion was done (e.g. adding the name of the sport) without human intervention (since the paper specifically touts ‘zero-shot’, I would imagine there is no human labeling required for this)

Please proof read, there are a few lurking typos/consistency issues:

-	 “While, their high-level label is barely recognized by actionrecognition models (e.g. such as Roundhouse kick in Kickboxing).”
-	SigLIP vs SigLip,and other model names such as simvlm in the references should be SimVLM
-	 similarly subaction vs sub-action, modeling vs modelling
-	A(i, m)  vs A(m,i) in eq 9 vs 12

---

> ### Author Response · Authors · 2025-09-29
> **Response to Reviewer Nrv2**
>
> We thank the reviewer for their positive assessment of our work. We also appreciate the thoughtful questions and suggestions, which have helped us clarify and strengthen the manuscript. Below we respond point by point.
>
> **1. On Cross-Modal Alignment with SigLIP**
> We appreciate the opportunity to clarify why a simple similarity function is meaningful in this context. The core design of modern image-language models is precisely to jointly train visual and textual encoders such that semantically related image-text pairs are mapped to nearby points in a shared embedding space. Concretely, SigLIP and related models are trained on hundreds of millions of image-caption pairs using a contrastive objective. This objective maximizes similarity for matched image-text pairs and minimizes it for mismatched ones, thereby aligning semantics across modalities. Thus, once trained, conceptually similar visual and textual content (e.g., a frame showing “dribbling” and the sub-action text “dribble the ball”) naturally produce high similarity scores.
>
> **2. On Sub-Action Prompt Variants and Sequence Sensitivity**
> We agree that understanding the effect of sub-action variability is important. In fact, our study explicitly investigates this through two prompt-generation strategies with different verbosity levels and sub-action sequence lengths:
>
> - **Short-fixed prompting:** concise, consistent-length sequences with minimal two- or three-word phrasings.
> - **Context-rich prompting:** longer, domain-aware sequences incorporating contextual elements in descriptions.
>
> The results of these two variants are reported in our ablation studies. Furthermore, we have **now added a qualitative example in the Appendix** that visually compares how the two prompt types lead to different alignment paths and highlights.
>
> **3. Additional Details and Zero-Shot Setup**
> We thank the reviewer for prompting us to include more implementation details. In the revised manuscript, we have:
>
> - **Expanded the background discussion on image-language models in Introduction:** clarifying why embeddings align across modalities.
> - **Added a more detailed description of signal smoothing in Section 3.5:** this uses a non-learnable 1D moving average over frame embeddings before alignment to improve temporal stability.
> - **Clarified how contextual expansion is performed fully automatically in Section 3.3:** domain names can be inferred from action name themselves without any human intervention or labeled data, preserving the zero-shot property. Including the domain name improves textual grounding.
>
> **4. Manuscript Quality and Minor Revisions**
> We appreciate the reviewer’s careful reading and helpful suggestions on presentation. We have conducted a thorough proofreading pass and corrected the issues mentioned. The changes should improve clarity and consistency throughout the manuscript.

---

### Review · Reviewer_UiFG · 2025-09-15

**Summary Of Contributions:**

The paper makes two primary contributions.

First, it introduces ActAlign, a novel framework that casts zero-shot fine-grained video classification as a sequence alignment problem, aligning LLM-generated sub-action scripts with video frames through Dynamic Time Warping. This reformulation enables a training-free, domain-general approach that preserves the open-set generalization of pretrained image-language models.

Second, the paper demonstrates strong empirical results on the ActionAtlas benchmark, showing that ActAlign significantly outperforms billion-parameter video-language models while using far fewer parameters, with ablation studies confirming the impact of temporal alignment and context-rich prompting.

**Audience:**

Yes

**Audience Explanation:**

Yes, this paper would likely interest TMLR’s audience. It tackles a rlevant problem and does so with a simple but clever idea: using LLM-generated sub-actions and DTW to add temporal structure to image-language models. The approach is lightweight, training-free, and interpretable, which makes it appealing compared to heavy video-language models.

**Claims And Evidence:**

Yes

**Claims Explanation:**

The paper’s main claims are generally well supported and accurate. The central idea of casting zero-shot fine-grained video classification as a sequence alignment problem using LLM-generated sub-actions and DTW is clearly implemented and validated.

Experimental results on the ActionAtlas benchmark back up the performance claims, showing significant gains over both CLIP-style baselines and billion-parameter video-language models, and the ablation studies confirm the contributions of temporal alignment and context-rich prompting.

The claims of interpretability and parameter efficiency are also substantiated through visualizations and comparisons. Overall, within the scope presented, the claims are credible and well supported.

**Requested Changes:**

The paper looks well executes and complete. However, some minor changes that could strengthen the study include:

1. Adding validation on an additional dataset beyond ActionAtlas to better support the claim of domain generality. Even a smaller benchmark (e.g., Something-Something, Kinetics subset) would strengthen the paper.

2. Provide a scalability discussion or analysis regarding runtime and memory cost when scaling to longer videos or larger class vocabularies, given the $O(NMT)$ complexity of DTW alignment.

3. Include more details and examples of the LLM-generated sub-action prompts and outputs (e.g., in the appendix). This would improve reproducibility and help readers understand how much performance depends on prompt design.

---

> ### Author Response · Authors · 2025-09-29
> **Response to Reviewer UiFG**
>
> We sincerely thank the reviewer for their thoughtful and encouraging feedback. Below, we address the suggested improvements point by point.
>
> **1. Domain Generality and Dataset Validation**
> We agree that broader validation would further strengthen the claim of domain generality. As noted in the paper, naive mean-pooling often suffices for coarse-grained actions (e.g., lifting or holding in something-something dataset) because they do not require sequential reasoning. Fine-grained actions, however, such as a hook shot in basketball, require following temporally ordered cues that pretrained image-language models usually do not capture and often are not represented in their training sets.
>
> We found ActionAtlas as a great benchmark to evaluate our approach as it spans over 50 diverse domains while maintaining a focus on fine-grained action granularity. To further support the claim of domain generality, we conducted additional analyses on the four largest domains within ActionAtlas. Across all of them, ActAlign consistently outperforms the SigLIP baseline.
>
> **2. Scalibility and Efficiency Discussion**
> We appreciate the request for deeper scalability discussion and have added detailed analysis in Appendix. Although DTW accounts for the majority of runtime, the computation is highly parallelizable: class-wise comparisons and video segments are independent and can be distributed across threads or GPUs.
>
> The end-to-end inference time for all 898 videos is ∼0.04 s/video on an AMD EPYC 7313 CPU (32 cores, 64 threads) without multi-threading.
>
> **3. Prompt Examples and Reproducibility**
> We fully agree that including richer examples of LLM-generated sub-actions would enhance reproducibility and clarity. To address this, we now provide representative alignmnet examples of both short-fixed prompting and context-rich prompting **in Appendix along with scripts**. They also show the relative expressiveness and impact of different prompting styles, helping readers assess how much performance depends on prompt design.

---

### Author Response · Authors · 2025-09-29
**Updated Submission and General Response to all Reviewers**

We thank all the reviewers for their detailed, constructive, and insightful feedback. It is encouraging that the reviewers found the idea elegant and easy to reproduce (pgw3), precise and well-explained (UiFG), and technically interesting with strong empirical evidence (Nrv2).

We have spent substantial efforts to further improve the clarity of the paper, including clearer explanation of cross-modal embedding alignment, normalization, and contextual expansion. We also **added new experiments** related to context-augmented mean pooling, bag-of-subactions, randomized sub-action order, and variance across multiple seeds. Additionally, we expanded the evaluation details following the ActionAtlas protocol and included new qualitative examples.

We have made all the requested changes and updated the manuscript. (Please note that the major changes are highlighted in **blue color** in the PDF.)

We look forward to any additional feedback and suggestions.

---

### Decision · Action_Editor_3iD8 · 2025-10-10

**Recommendation:** Accept as is

**Audience:**

Yes

**Audience Explanation:**

The proposed zero-shot video classification method, ActAlign, formulates video classification as a sequence alignment problem. The approach demonstrates promising performance on the ActionAtlas benchmark, making it a valuable and interesting contribution to the research community.

**Claims And Evidence:**

Yes

**Claims Explanation:**

To address the challenge of zero-shot fine-grained video classification, this work introduces ActAlign, a framework that formulates video classification as a sequence alignment problem. Specifically, for each action class, a large language model (LLM) decomposes the action into an ordered sequence of sub-actions. A frozen image-text model (SigLIP) is then employed to embed both video frames and sub-actions, which are aligned using Dynamic Time Warping (DTW). The proposed ActAlign method achieves promising results on the ActionAtlas benchmark.

All three reviewers recognized the effectiveness of the proposed approach and found the claims well-supported by empirical evidence. Some concerns were raised regarding evaluation details, inclusion of additional baselines and ablations, and writing clarity.

The authors’ rebuttal effectively addressed these concerns. Ultimately, all three reviewers recommended acceptance. After reviewing the submission, reviews, and rebuttal, the action editor concurred with the reviewers and recommended acceptance of the paper.